# Emotional Responses to the Visual Patterns of Urban Streets: Evidence from Physiological and Subjective Indicators

**DOI:** 10.3390/ijerph18189677

**Published:** 2021-09-14

**Authors:** Zijiao Zhang, Kangfu Zhuo, Wenhan Wei, Fu Li, Jie Yin, Liyan Xu

**Affiliations:** 1College of Architecture and Landscape Architecture, Peking University, Beijing 100871, China; zjzhangpku@foxmail.com (Z.Z.); zhuokf21@mails.tsinghua.edu.cn (K.Z.); wenhanwei@pku.edu.cn (W.W.); lifu93@pku.edu.cn (F.L.); 2College of Architecture and Urban Planning, Tongji University, Shanghai 200092, China

**Keywords:** urban street, visual pattern, environmental perception, virtual reality, physiological monitoring, subjective evaluation

## Abstract

Despite recent progress in the research of people’s emotional response to the environment, the built—rather than natural—environment’s emotional effects have not yet been thoroughly examined. In response to this knowledge gap, we recruited 26 participants and scrutinized their emotional response to various urban street scenes through an immersive exposure experiment using virtual reality. We utilized new physiological monitoring technologies that enable synchronized observation of the participants’ electroencephalography, electrodermal activity, and heart rate, as well as their subjective indicators. With the newly introduced measurement for the global visual patterns of the built environment, we built statistical models to examine people’s emotional response to the physical element configuration and color composition of street scenes. We found that more diverse and less fragmented scenes inspired positive emotional feelings. We also found (in)consistency among the physiological and subjective indicators, indicating a potentially interesting neural−physiological interpretation for the classic form−function dichotomy in architecture. Besides the practical implications on promoting physical environment design, this study combined objective physiology-monitoring technology and questionnaire-based research techniques to demonstrate a better approach to quantify environment−emotion relationships.

## 1. Introduction

The inquiry on the relationship between people and the built environment reflects a long-established intellectual tradition [1,2,3,4]. Indeed, more than 55% of people now live in cities [5], with others visiting the city occasionally. Ever since the early-mid 20th century, classic research has revealed that the sociological and symbolic meaning of space, as well as people’s experience of it, would strengthen their perceptions of places [1,3]. Therefore, a high-quality design of physical settings should conform to human sensory characteristics so as to promote positive psychological responses [6]. Specifically, as one of the most common forms of the built environment, urban streets gather a large number of human activities, and their functional diversity, façade forms, and scale characteristics have impacts on the environmental perception and behavior of the space users [7,8]. Meanwhile, with the progress of cognitive science, scholars have increasingly realized that the human−built environment relationship has deep psychological roots [9]. As a result, deriving from the classic, phenomenological paradigm, which is mainly based on subjective experiences and feelings from a macroscopic perspective, a new research paradigm has emerged in the last decade. It focuses on places, spaces, and the individual feelings amid them, and uses quantitative methods to explore the impact of physical settings on human perception in a logical positivist manner. This paradigm has become the basis for the emerging school of evidence-based design [10,11].

In empirical environmental psychology research, scholars generally conceptualize human-environmental perception and behavior processes with an exposure−cognition−behavior response framework [12]. This framework enumerates three major phases, namely (1) environmental exposure on human, such as visual stimulus or noises; (2) human cognition of the exposure, which is mainly affected by individual differences in age, gender, social, adaptation ability, etc.; and (3) human behavior responses, including emotional arousal, stress relief, or appraisal to the current situation. The conceptualization stresses the importance of accurate measurements of the exposure and behavior in empirical research. Visual exposure, as the most direct sensory approach [13], has drawn particular research attention. It has been revealed that people’s visual exposure to the built environment affects their subjective perceptions and emotions, such as comfort, safety, vitality, and depression [14,15,16,17,18]. Specifically, regarding the source of visual exposure, natural elements have become a prominent research subject. A large number of studies in the field of environmental psychology and landscape architecture have proven that exposure to the natural environment can help reduce stress and promote well-being [10,19,20,21,22,23]. However, compared with the intensive attention on natural elements, the research progress of environmental perception on artificially built elements appears not commensurate to date. A few recent works revealed that some artificial elements of streetscapes, such as the proportion of windows on the street, the proportion of active street frontage, and the number of pieces of street furniture, have significant effects on pedestrian activity [24]. Moreover, they found that both entropy (aggregate architectural variation) and building height of the urban streetscape could affect restoration likelihood for citizens [25]. Although the natural−artificial dichotomy is a convenient technical treatment, both types of elements usually present simultaneously in the real world and make cognitive impacts interdependently. Thus, an integrated perspective is necessary.

Furthermore, although the research on the impacts of specific artificial environment elements on human emotions has become essential, the “comprehensiveness” principle of Gestalt Psychology suggests that the effects of specific elements do not replace that from the whole environment [26]. Indeed, many long-established environmental design principles, such as symmetry, rhythm, balance, and coordination, all concentrate on the holistic impression of the environment. The subtleness in them cannot be reduced to partial, element-wise measurements such as the count, area, and ratio of certain elements. Although research on the latter, as reviewed above, has progressed to some extent, the impact of the comprehensive visual form of the environment on human perception has not been fully examined yet.

Under the exposure−cognition−behavior response framework, we believe the reasons for the existence of the aforementioned research gaps lie in both exposure and behavior response (perception evaluation) procedures. First, in the exposure procedure, on the one hand, the traditional way of exposure assessment in lab-based experiments usually involves photo-showing to the participants [27]. With the limited angle of view of those photos, the exposure approach has an intrinsic limitation in terms of representing the holistic environment. On the other hand, field-based experiments face challenges in controlling physical environmental confounders. Meanwhile, for the auditing of the environmental exposure, the problem is that the “comprehensive” measurements of the environment are not as straightforward as the element-wise measurements. For example, concepts such as “rhythm” or “balance” are difficult to measure objectively because different individuals may understand them in quite different ways. Therefore, although accurate “comprehensive” measurements of the environment are necessary, progress so far is still limited. Next, in the procedure of perception evaluation, interviews and questionnaires are the traditional methods for data collection [1,28,29]. Psychological scaling methods are commonly used for evaluating perception [30,31]. These methods, representing the stated preference approach, inevitably introduce respondents’ subjective interference when answering questions, and hence have negative impacts on accurately measuring their true feelings [32].

The recent development of new technologies has shed light on these limitations, but also brings about new challenges. First, the introduction of virtual reality (VR) technology, together with the panoramic images, would allow for a more realistic representation of the environment and thus enable immersive exposure in lab-based experiments. VR-based research has been increasingly applied to environmental perception experiments [16,18,31]. Research has revealed that compared with photos, VR exposure is closer to the physical environment in terms of psychological and physical responses [33], and awakens participants’ emotions better [34]. However, a technical issue is that most panoramic photo-based research has hardly considered the distortion of spherical panoramic images in the process of environmental auditing, resulting in a bias in some vital measurements such as area sizes and angles. Some studies used projection transformation to obtain more accurate image auditing results, such as generating fisheye images towards the sky through projection transformation [35]. However, there is about 50% of information loss as the part of the image under the horizontal plane of the viewpoint is ignored. Therefore, appropriate photogrammetric processing should be performed to reduce these auditing errors. Next, with the recent development of portable physiological sensors, some human cognitive states and feelings can be captured non-intrusively. These measurements include electroencephalography (EEG), electrodermal activity (EDA), and heart rate (HR) [36,37]. They have been proven to be relevant and accurate indicators for people’s emotions, and are convenient to obtain in VR environments [38,39]. To some extent, they constitute a means to observe the revealed preference (RP) of the respondents. Moreover, they can avoid the measurement bias caused by differential individual behavioral patterns and have been widely used in relevant research accordingly [30,31,37]. However, the relationship between some of these “new” perception evaluation measurements and the “old”, self-report measurements remains unclear [40]. In addition, the consistency among those “new” measurements and the fact that existing research commonly observes only one type of the “new” measurements have hindered the cross-validation and comparative analysis of the effects of different types of “new” measurements, rendering the above questions yet to be answered.

In sum, despite the recent considerable progress in the field of environmental perception, the mechanism of how built environment design affects our emotions has not yet been thoroughly revealed [41]. Research gaps still exist, particularly in addressing the following questions: (1) What are the auditing methods and techniques for the comprehensive examination of the built environment? (2) What is the relationship between the “new”, non-intrusive perception evaluation measurements and the “old”, interview- or questionnaire-based ones? (3) Is there consistency among different types of the “new” measurements? To address these questions, this study investigates the effects of the comprehensive visual pattern of urban streets on human emotions. By utilizing panoramic image-based VR techniques, we conducted lab-based cognitive experiments, where participants were exposed under controlled conditions and had an immersive experience in the test environments. In the experiment, we simultaneously used the “traditional” interview- and questionnaire-based method and the “new”, non-intrusive methods to obtain the participants’ subjective and physiological responses to the exposure, respectively. We conducted the “comprehensive” audition of the environment by introducing global landscape metrics representing the spatial configuration and color composition of the scenes, as well as an equal-area projection process that ensures the auditing results are accurately in line with the real proportion in visual perception. We then built statistical models to analyze the impact of the comprehensive environment features on the participants’ emotional responses. Finally, we examined the differences among response measurements to evaluate their consistency and discussed the possible explanation and implication from a cognitive science perspective.

## 2. Methods

### 2.1. Study Design

We conducted a VR-based experiment in which participants were exposed to urban street scenes, which were collected on-site in Beijing using a dual fisheye panoramic camera (RICOH THETA V, Ricoh Company Ltd., Tokyo, Japan) on a clear day. These scenes reflect a variety of spatial scales (Figure 1; all scenes are available upon request). The spatial scales of the streets can be divided into three categories: small (two lanes or less), medium (four to six lanes), and large (six lanes or more). There were 39 selected urban street scenes in total, which were randomly assigned to four groups, with 12 scenes in each group and some of them repeated in different groups. All participants were randomly assigned to explore one of the four groups during the experiment. The physiological indicators of the participants, including EEG (Emotiv EPOC+, EMOTIV Inc., San Francisco, CA, USA), EDA, and HR (E4 wristband, Empatica Inc., Cambridge, MA, USA), were obtained by bio-monitoring sensors in real-time, and the subjective evaluations were conducted through a question-and-answer interview.

We used the convenient sampling method to enroll as many participants as possible from May to June in 2019. By posting the brief information of this experiment online, we recruited 30 healthy adults to participate in the experiment. Due to the malfunctioning of the devices or poor data quality, the data of four participants were ruled out. The data of 26 participants were accepted into the analysis phases. All qualified participants were students from Peking University and had voluntarily signed up for the experiment. The study complies with the Declaration of Helsinki and the ethics rules of Peking University.

### 2.2. Visual Patterns of the Urban Street Scenes

#### 2.2.1. Semantic Segmentation of the Images

Auditing the environmental elements is a key step in quantifying environmental exposure. Image analysis algorithms based on machine learning, including semantic segmentation, emotional evaluation, scene perception, scale measurement, etc., have been increasingly used in image-based visual auditing [15,42,43]. Among them, semantic segmentation technology using the graph convolution algorithm can produce accurate results on different types of images with consistent standards, and has become a frequently used method in urban environmental research to evaluate the visual quality of urban space [44,45]. We applied the ResNet semantic segmentation package to process the collected images of the street scenes and used the MIT ADE20K scene parsing dataset as the basic training set [46]. Released by the MIT Computer Vision team, the set is the largest open-source dataset for semantic segmentation and scene parsing. The constituent elements of the scenes are divided into 150 classes, such as plants, sky, and roads [46] (Figure 2).

#### 2.2.2. Color Classification of the Images

As an important element of visual perception, color is usually quantitatively measured and evaluated using color models [47]. To simplify the complex color characteristics of the scenes, we grouped the colors in the street scenes into easily identifiable categories for further analysis. The color classification in this study is based on the HSV color model, which is close to the way the human visual system perceives and describes colors, and is widely used in image processing and expression related to human color perception [47]. The color parameters in this model are hue (H), saturation (S), and value (V). Based on the specific H, S, and V threshold rules (see Appendix A), we categorized the color of each pixel in the street scene images into six classes: red, green, blue, gray, white, and black. The results of the color classification show that the shapes of the objects in the original image are well retained, and important constituent elements such as the sky, roads, buildings, and vegetation, are clearly distinguished (Figure 2). Therefore, the subsequent analysis based on this will reflect the color composition in the real scene well.

#### 2.2.3. Projection Transformation of Panoramic Images

As the original panoramic images were spherical and were by default output using the Equirectangular Projection in which the image resolution ratio was 2:1, they looked “distorted”, in that the area of high-latitude elements were too large and the directions were distorted. To retrieve the accurate area of visual elements, we used the Hammer Projection, one of the commonly used equal-area projection methods in cartography, to conduct projection transformation on the panoramic images in the auditing process (Figure 3).

#### 2.2.4. Visual Pattern Indicators of the Street Scenes

To define the proper indicators for the global visual pattern of the street scenes, we borrowed from the concepts of the landscape pattern metrics in landscape ecology [48]. By analogizing the elements (in semantically segmented or color-classified images) in a street scene to land patches in a landscape, element categories such as land patch classes and the scene as the entire landscape, the visual pattern of a street scene can be measured by using the same techniques to measure landscape patterns. Per the analogies above, the visual pattern metrics here included patch (element-wise), class (element category-wise), and landscape (scene-wise) metrics, in which the class and landscape metrics quantified the overall or “global” composition and configuration of a scene with highly condensed information [48]. Technically, we chose three metrics to measure the multi-dimensional visual structure characteristics: percentage of landscape (PLAND) at the element category level, and landscape division index (DIVISION) and Shannon’s diversity index (SHDI) at the entire street scene level (Table 1). Specifically, PLAND measures the proportion of each element category in the street scene; DIVISION measures the degree of fragmentation of the scene, i.e., the percentage occupied by relatively small, or “fragmented” visual elements in the scene, with a higher DIVISION value indicating increased fragmentation; and SHDI measures the degree of diversity of the scene, i.e., the richness of different visual element types presented in the scene, with a higher SHDI value indicating a more equitable areal distribution of the visual element types [49]. We used FRAGSTATS (a spatial pattern analysis program for categorical maps) to compute the metrics [50].

### 2.3. Measurements of the Participants’ Environmental Perception

We measured the participants’ emotional response through two approaches: objective indicators and subjective indicators [51]. For the former, electroencephalography (EEG) measures human brain function and activity. EMOTIV EPOC+ can further generate emotional indicators in real-time based on brain signals, which are used to explore the emotional changes as a response to different environmental settings, such as urban and natural environments [30,51,52,53]. In our experiment, EEG data were obtained from the EMOTIV EPOC+, a mobile EEG headset that has 14 channels to capture brain wavebands. The metrics calculated by the EMOTIV performance metrics algorithms for cognitive states were used in this study. They were output at 0.1 Hz and included the following six metrics: engagement (En), excitement (Ex), stress (St), relaxation (Re), interest (In), and focus (Fo). According to the official user’s manual, the definitions of the metrics are as follows [54]:Engagement measures “the level of immersion in the moment and is a mixture of attention and concentration and contrasts with boredom”;Excitement is “an awareness or feeling of physiological arousal with a positive value”;Stress is “a measure of comfort with the current challenge”, with “a low to moderate level improving productivity”, and “a high level tending to be destructive”;Relaxation is “a measure of an ability to switch off and recover from intense concentration”;Interest measures “the degree of attraction or aversion of current stimuli”, with high scores “indicate a strong affinity”, and low scores “indicate a strong aversion to the task”;Focus is “a measure of fixed attention to one specific task”.

We used a scaled value ranging from 0 to 1 for these six EEG performance metrics according to the official equation (based on directives on https://emotiv.gitbook.io/emotivpro-v2-0/csv-files#performance-metrics-data, accessed on 6 September 2021) to make it more comparable among samples.

Other objective physiological indicators that concern people’s emotional and cognitive states include EDA, HR, etc., and are often used to measure some specific emotional states, such as stress level [31,55] and engagement [56]. The Empatica E4 wristband was used in our experiment, which is a wearable device that captured wearers’ EDA at 4 Hz and HR derived from the blood volume pulse at 64 Hz. EDA reflects the autonomic sympathetic arousal integrated with emotional and cognitive states [57,58], whose signals can be characterized by a slowly varying tonic activity (i.e., skin conductance level (SCL)) and a fast varying phasic activity (i.e., skin conductance responses (SCRs)) [59]. As SCRs are sudden rises in the skin conductivity in response to an increase in sympathetic activity, they are often used to measure event-related phasic changes linked to emotional stimuli [60], in which SCR shows a steep incline to the peak and a slow decline to the baseline [60]. After correction for the motion artifact [61], we used the Ledalab tool in MATLAB to decompose the EDA data streams into SCL and SCRs [59,62]. As the immersion time of each scene in our experiment was relatively short, we used two indicators—the number of SCRs (SCR.n) and the sum of SCR amplitude (SCR.Amp)—in each scene to measure the participants’ emotional arousal. HR reflects a combination of sympathetic and parasympathetic activity and is also an indirect measurement of emotional states [40]. We used the average value of HR (HR.Avg) directly generated by E4 to measure the difference in the average level between the scenes.

For the subjective indicators, to better reflect the participants’ environmental feelings for different scenes affected by the visual quality, we used three indicators to measure the participants’ subjective evaluation of the scenes, interestingness, comfort, and vitality, which refer to the indicators measuring experiential qualities of urban public spaces in previous research [63,64]. Interestingness focuses on the likeability and imageability, which could induce positive feeling and a connection to the place; comfort comes from walkability, seating provision, and greenery, and is thus an evaluation of the basic physical service-provision function of the place; and vitality describes whether the place is vibrant, playful, animate, and, especially in the Chinese context, has a sense of livelihood. Each indicator contains three words that represent negative, neutral, and positive feelings, respectively, where “impassive” means that the participants do not have the obvious tendency or related feelings on this indicator (Table 2). During the experiment, the participants were provided with three options for each question (e.g., “What do you think of this scene, insipid, interesting, or impassive?”) and were asked to choose the one closest to their feelings. The choices were based on the overall emotional evaluation of the scene, which had been explained to the participants before the experiment. Then, the options were transformed into ordinal scores from one to three, with higher scores indicating more positive feelings (Table 2).

### 2.4. Experiment Procedure

All of the experiments were conducted in a laboratory at the College of Architecture and Landscape Architecture, Peking University. The experiment included three stages: preparation, exposure, and review (Figure 4). In the preparation stage, the participants were informed of the procedure and precautions of the experiment. Then, they wet their hair with conductive liquid and wore the Emotiv EPOC+ headset, the Empatica E4 wristband, and the HTC Vive VR headset with the researchers’ help. After the experimental equipment was calibrated, the participants were given a 3-min break with their eyes closed.

In the exposure stage, the participants were randomly assigned to experience one of the four groups of scenes. They were told to open their eyes when the first scene started to screen in VR, and to look around and observe the scene for 30 s. Then, they were asked three questions about their subjective evaluation of the scene (i.e., interestingness, comfort, and vitality). After answering the questions, they were told to close their eyes and rest for 30 s to restore to a stable level of physiological states, and then entered the next scene. Time nodes of each step were marked on the bio-monitoring sensors.

After all the scenes had been experienced, all devices were removed from the participants. Then, a brief interview was conducted with the participants as a review. The participants were asked about their most impressive scenes and the reason and then spoke freely about anything they felt of the scenes. Then they completed an online survey about their demographic information (e.g., age, gender, and academic degree). The whole experiment lasted around 35 min.

### 2.5. Statistical Analysis

We analyzed the effects of urban streets’ visual patterns on human emotions from two dimensions: the constituent element patterns and the color patterns. However, there was a high correlation between PLAND of some types of constituent elements and the same for some types of colors. To avoid multicollinearity, only the indicators that had a lower correlation and a strong impact on the visual pattern were kept in the models. The final independent variables for the constituent element pattern included PLAND of plant, PLAND of the sky, PLAND of road, DIVISION of constituent elements (DIVISION of CE), and SHDI of constituent-elements (SHDI of CE); those for the color pattern included PLAND of green-class, PLAND of blue-class, PLAND of red-class, DIVISION of colors (DIVISION of CO), and SHDI of colors (SHDI of CO). All visual pattern indicators were continuous variables.

We also added the street scale variable and gender of the participants to the models as control variables. Each type of street scale was defined as a dummy variable, with the medium street scale as the reference, as it is most common in cities. The gender of the participants was also defined as a dummy variable, where male equals 1 and female equals 0. Lastly, to avoid any potential overfitting issue caused by too many variables entering the model, we fit the models for the constituent elements and the colors separately—this also helped avoid multicollinearity as the variables in the two groups tended to correlate with each other. Thus, for the physiological indicators, we built two groups of multiple linear regression models to test the effects of the visual pattern variables on them. Formulas (1) and (2) describe model groups 1 and 2, respectively. All models were fit using the IBM SPSS Statistics 26 statistical package.

This is example 1 of an equation:(1)$$\begin{array}{cc} Y=\beta_{0}+\beta_{1}PLAN & D\;of\;plant+\beta_{2}PLAND\;of\;sky+\beta_{3}PLAND\;of\;road\\ & +\;\beta_{4}DIVISION\;of\;CE+\beta_{5}SHDI\;of\;CE\\ & +\;\beta_{6}large\;street\;scale+\beta_{7}small\;street\;scale+\beta_{8}gender\\ & +\;e_{i} \end{array}$$
(2)$$\begin{array}{cc} Y=\beta_{0}+\beta_{1}PLAN & D\;of\;green-class+\beta_{2}PLAND\;of\;blue-class\\ & +\;\beta_{3}PLAND\;of\;red-class+\beta_{4}DIVISION\;of\;CO\\ & +\;\beta_{5}SHDI\;of\;CO+\beta_{6}large\;street\;scale\\ & +\;\beta_{7}small\;street\;scale+\beta_{8}gender+e_{i} \end{array}$$
where Y = EEG, SCR, and HR indicators.

For the subjective evaluation indicators, we built two groups of multiple logistic regression models to test the effects of the visual pattern variables on them. Formulas (3) and (4) describe model groups 3 and 4, respectively.
(3)$$\begin{array}{cc} {\mathrm{LogitP}}_{n}=\beta_{0}\;+\;\beta_{1} & PLAND\;of\;plant\;+\;\beta_{2}PLAND\;of\;sky\;+\;\beta_{3}PLAND\;of\;road\\ & +\;\beta_{4}DIVISION\;of\;CE+\beta_{5}SHDI\;of\;CE\\ & +\;\beta_{6}large\;street\;scale\;+\;\beta_{7}small\;street\;scale\;+\;\beta_{8}gender\\ & +\;e_{i} \end{array}$$
(4)$$\begin{array}{cc} {\mathrm{LogitP}}_{n}=\beta_{0}+\beta_{1} & PLAND\;of\;green-class+\beta_{2}PLAND\;of\;blue-class\\ & +\;\beta_{3}PLAND\;of\;red-class+\beta_{4}DIVISION\;of\;CO\\ & +\;\beta_{5}SHDI\;of\;CO+\beta_{6}large\;street\;scale\\ & +\;\beta_{7}small\;street\;scale+\beta_{8}gender+e_{i} \end{array}$$
where ${\mathrm{LogitP}}_{n}=\log[P\left(n\right)/P\left(1\right)]$; *n* = 2 or 3 taking 1 as reference.

Furthermore, we conducted correlation analyses to test the relationships between different indicators. Among them, the Pearson correlation coefficient was used between the continuous variable indicators, that is, the physiological indicators. The Spearman correlation coefficient was used between the subjective evaluation indicators themselves and between the subjective evaluation indicators and the continuous variables, as subjective evaluation indicators were ordinal (Figure 5).

## 3. Results

### 3.1. Descriptive Statistics of the Participants, Scenes, and Perception Measurements

The descriptive statistical characteristics of the participants, scenes for exposure, and outcome measurements are presented in Table 3. The gender ratio of the 26 participants was balanced. Among the constituent element characteristics of the 39 selected scenes, the average proportions of plants, sky, and road were 16.62% (±10.05%), 18.75% (±7.16%), and 40.09% (±3.58%), respectively, generally reflecting the visual perception of urban streets. The proportion of roads accounted for nearly half of the view and had small fluctuations. The proportion of the two natural elements (i.e., plants and sky) accounted for less than 20% on average and had large fluctuations. Among the color characteristics, the average proportions of green-class, blue-class, and red-class colors were 5.30% (±4.74%), 15.37% (±9.67%), and 5.75% (±3.75%), respectively. Their proportions were relatively low and fluctuate greatly, indicating that most of the people’s sight in urban streets is still occupied by low-saturation colors like black, white, and gray.

There were four types of outcomes produced by the experiment: EEG, SCR, HR, and subjective evaluation. The total sample size of the experiment is the sum of the product of the number of participants and the number of scenes each participant experienced. Some invalid data were generated due to the malfunction of devices during the experiment. After data cleaning, the numbers of valid EEG, SCR, HR, and subjective evaluation were 208, 213, 310, and 310, respectively.

### 3.2. Results of the Statistical Models for the Physiological Indicators

The effects of the street visual pattern characteristics, street scale, and gender of the participants on the physiological indicators, including EEG, SCR, and HR measurements, are shown in Table 4 and Table 5. For the EEG models, first of all, only the models for “excitement” (EEG.Ex) and “interest” (EEG.In) were held at a statistically (marginally) significant level, while the four models did not. Next, most visual pattern independent variables were not statistically significant, except the PLAND of blue-class for the EEG.Ex model, which was only marginally significant (at *p* = 0.108). On the contrary, as a control variable, gender had a significant impact on all of the four models for EEG.Ex and EEG.In, and suggests that the female participants’ interest is higher while excitement is lower compared with the male participants overall. 

For the SCR models, both the number of SCRs (SCR.n) and the sum of SCR amplitude (SCR.Amp) could be explained at a statistically significant level. The participants’ gender once again had a significant impact, and males had an overall higher level of those indicators than females. In addition, the proportion of roads had significant positive effects on the sum of the SCR amplitude. 

The models for HR were not held at a statistically significant level.

### 3.3. Results of the Statistical Models for the Subjective Evaluation Indicators

The effects of street visual pattern characteristics, street scale, and gender of the participants on subjective evaluation indicators are shown in Table 6 and Table 7. Overall, among the three indicators, comfort and vitality could be explained at a statistically significant level, while the interestingness model was not statistically significant.

Through the comparison of the two groups of influencing factors, we found that the natural elements in the street view and the proportion of blue-class and green-class colors had significant impacts on people’s subjective feelings. An increase in the proportion of plants significantly improved the comfort rating and made people feel relaxed. In addition, the proportion of green-class colors had positive effects on both comfort and vitality ratings at various degrees. Regarding vitality, the proportion of sky for “dull” scenes had higher evaluation scores than that for “impassive” scenes, while “vibrant” scenes had a higher proportion of blue colors, as well as a higher proportion of roads than “dull” scenes.

The overall visual pattern and the street scale also had significant impacts on people’s subjective feelings. For comfort rating, “depressive” scenes had a higher fragmentation level and a lower diversity level of the constituent elements and colors. In addition, “relaxed” was more likely to be chosen in scenes with a large street scale. For vitality ratings, “dull” scenes also had a higher fragmentation level and a lower diversity level of the constituent elements, while the visual pattern of colors had no significant impact, which means people’s feelings about vitality are more sensitive to the constituent elements than colors. In addition, vitality was scored higher on the large street scale, which is similar to the comfort rating. Besides, unlike the physiological indicators, participants’ gender had no significant effect on the subjective evaluations.

### 3.4. Correlation of Different Types of Perception Results

Most of the correlations between the 12 perception indicators obtained from the four different measurement approaches in the experiment were not statistically significant. Only four pairs of measurements were closely related (with a correlation coefficient greater than 0.4), which are: (1) EEG.In and EEG.Fo (0.400 ***); (2) SCR.n and SCR.Amp (0.411 ***); (3) EVA.In and EVA.Vi (0.436 ***); and (4) EVA.Co and EVA.Vi (0.561 ***). It is noteworthy that all pairs were obtained using the same sensing approach. 

If we lower the standard for a “close” relationship to include all statistically significant correlations, then a couple of new pairs joined, in which three pairs were obtained using different sensing approaches: EEG.Ex was positively correlated with SCR.n (0.161 **), and negatively correlated with EVA.Co and EVA.Vi (−0.198 ***, −0.179 ***). There was no significant correlation between the subjective evaluation measurements and the SCL and HR measurements (Table 8).

## 4. Discussion

### 4.1. Effects of Visual Patterns on People’s Emotions

The findings show that the proportion of vegetation/greenish colors in urban street scenes would significantly and positively affect various human emotions, including the comfort and vitality feelings, which is consistent with the findings in previous research on the stress-reduction and restorations effects of vegetation [20,23]. Moreover, our findings further show that the global visual pattern, including the spatial configuration of street elements and color composition of the scenes, would also affect human emotions, which was not fully discussed in previous research. Specifically, more diverse and less fragmented street scenes are found to be associated with positive emotions. Methodologically, our findings suggest that the global visual pattern auditing of street scenes is necessary because it empowers the capturing of the overall structural information of the environment in a more comprehensive way than traditional, element-wise auditing methods, and thus is recommended in future environmental perception research. 

Meanwhile, our findings also have obvious urban planning and design implications: planners and designers should avoid too uniform physical elements and color structures in urban street design. They should also try to reduce the fragmentation of street scenes to bring about more positive feelings for users of the streets. In some sense, we may view these implications as a psychological testimony of the traditional design know-hows, as they are generally consistent with the latter’s emphasis on the importance of symmetry, rhythm, balance, and coordination in architecture and city design [65,66]. Beyond that, our findings even go further and uncover more subtle relationships between design methods and human emotions. For example, we show that people are more sensitive to the element configuration than color composition in street scenes. Overall, the new findings would thus help planners and designers to “control” their plans in a more accurate way for better emotional effects. 

### 4.2. Improvement on Image Auditing Methods

Technically, our research shows the importance of proper image processing in the auditing of the environment. Generally speaking, for the inevitable optical and geometrical distortion in the process of imaging, the environmental perception research that utilizes images of all kinds, especially those whose results rely on the accurate auditing of the images, should perform optical and photogrammetrical corrections and projective transformation of the images whenever it is appropriate for the research goals [67]. For example, as this research deeply relies on the accurate auditing of area sizes of the elements in the image, an equal-area projection transformation is necessary. However, for research that focuses more on the relative directions of the elements in the images, an equal-angle projection within a certain field of view would be more appropriate.

### 4.3. Consistency of Different Types of Environmental Perception Indicators, and Some Methodological and Theoretical Reflections

Lastly, and most importantly, one innovation in our research is the realization of synchronized measurements of the four types of physiological and subjective emotional indicators within the VR-based experiment framework. We hypothesized that there would be a significant level of consistency of results among the different types of indicators as they are responses to the same exposure. However, the results, in general, do not support our hypothesis. In particular, the physiological results measured by the EEG, SCR, and HR are largely uncorrelated with each other, and they both show limited correlation with the results from subjective evaluations. Although inconsistency may be partly attributed to technical reasons, such as measurement errors caused by poor electrode contact, or noise signals caused by the discomfort from wearing the devices or from winding wires, we can still make some interesting conjectures from the perspective of the neural mechanism of human emotions.

Although the EEG-based emotional indicators have been proven effective in detecting variations in human emotions in various types of the environment [51,53,68], we only found statistically significant results on two measurements: excitement and interest. It is noteworthy, however, that among the six “emotional” indicators provided by the device, only these two have a straightforward neural mechanism, while those for the other four are relatively complicated. Specifically, excitement is associated with physiological arousal and is an autonomous conditioning reflex activity. It is caused by the activation of the sympathetic system and is usually accompanied by physiological responses including sweating and heart rate increasing [54]. In our experiment, excitement (EEG.Ex) does have a positive, but weak correlation with sweating (SCR.n). As the correlation is no surprise, its weakness may be caused by the fact that there lacks significant variation among the street scenes, such that the effects are not as prominent as those found between more drastically different settings, such as nature vs. artificial environments. On the other hand, our results show a negative correlation between EEG.Ex and the two subjective evaluation results. This is proof that even with the “modern”, direct measurement methods for people’s physiological status, the “traditional”, interview- or questionnaire-based subjective evaluation methods are still necessary at least for the autonomous conditioning reflex activities, as they give the direction of the arousal, which is somewhat hard to obtain with the “modern” sensors at their current technological state.

On the contrary, interest reflects the valence of emotions [54]. Being an indicator for people’s attraction or aversion to current stimulus [54], interest is an evaluative conditioning reflex activity [69]. Different from autonomous conditioning such as excitement which is usually considered to be associated with “basic”, physiological activities, evaluative conditioning is more related to people’s “advanced”, psychological activities such as aestheticizing. Such dichotomy naturally reminds us of another dichotomy in architecture, the one between function and form of the physical environment. For example, in Vitruvius’ famous three principles for architecture, firmitatis (stability) and utilitatis (utility) concern the functional dimension, while venustatis (beauty) concerns the form−aesthetical dimension [70]. One cannot help asking that whether there are resonations between the two dichotomies. This is of course a complicated question, and particularly, even if we can relate the “basic” physiological-rooted emotions such as excitement with the functional dimension of architecture, we still have very limited knowledge in that between the “advanced” psychological valence and the form dimension of architecture. Although our experiment cannot give a direct answer to the above question, it does imply our inability to answer it with the currently available research techniques. More advanced physiological monitoring equipment, such as fMRI [71], may help answer the question. However, under its current state of technology, it is impossible to integrate fMRI monitoring with various other monitoring devices to realize a multi-dimensional synchronized observation of human emotions as we did in our experiment. We thus hope that future technological progress may equip us with more portable and accurate devices. 

Methodologically, the above discussion indicates both merits and limitations in introducing the direct physiological monitoring technology in the research of the environment−emotion relationship, and the need for incorporating qualitative analysis alongside the quantitative calculations. The new technology does help reveal different pictures of people’s emotional response to the environment, but the inconsistency among different measurement approaches, as well as our limited mechanistic understanding regarding the inconsistency, reminds us of the potential risk if we rely too much on the new technology. In contrast, the “traditional”, interview- or questionnaire-based approaches, although they do not directly touch upon the physiological roots of emotions, are proven valid, effective, and robust [72]. On balance, currently, a combination of the two approaches in the research of the environment−emotion relationship could be a reasonable approach to uncover the hidden relationship among dimensions of physical environment design, human emotions, and their physiological basis.

Moreover, the discussion also triggers interesting theoretical reflections on the essence of the human—(built) environment relationship. Notwithstanding possible technical errors, questions remain as to why most of the “common” EEG data measurements do not correlate to the respondents’ expressed emotions. Following Eco’s argument [4], does this mean that there are unknown emotional modes people have to decipher the semiotics of architecture? Or, following Merleau−Ponty’s viewpoint [73], does this mean that there are unknown mechanisms people follow to experience the built environment and construct the meaning of places? The modes and mechanisms, if uncovered, may help us establish a more direct relationship between human emotions and their neural−physiological roots. They would be also beneficial for deepening our understandings on the human−(built) environment relationship from both logical positivist and phenomenological paradigms.

## 5. Conclusions

In this paper, we scrutinize people’s emotional responses to various urban street scenes with a VR-based, immersive exposure experiment with 26 participants. We applied a refined method for more accurate visual pattern auditing for urban street scenes. We also conducted a synchronized observation of the participants’ EEG, EDA, and HR indicators, as well as their subjective evaluation results. We found that the two global visual pattern measurements, the physical element configuration and color composition of street scenes, significantly affect people’s environmental perception in that more diverse and less fragmented scenes would introduce positive emotional feelings. This finding may be of help to planners and designers for a deeper understanding of the psychological impacts of the physical urban street environment, and thus promote better designs. Besides, an analysis on the consistency among the physiological and subjective indicators shows subtleness in such relationships and implies the possible neural−physiological basis for the traditional form-function dichotomy in architecture. This finding suggests that the combination of the new, direct physiology-monitoring technology and the “traditional”, interview- or questionnaire-based research techniques in the research of environmental perception would help better unveil the mechanisms of the environment-emotion relationships.

Our study has a few limitations. First, participants were young, healthy students from Peking University. In that case, the generalizability of the results is limited. We plan to recruit a larger and more diverse population in our future experiments. Second, by using VR and panoramic images, we provided our participants’ immersive experience of exposing to virtual urban street scenes. However, this method could only provide a visual connection with the environment and ignored other sensory approaches such as auditory and olfactory. With the fast development of VR and related technologies, we plan to initiate some VR-based environmental perception research looking at audio−visual interactions. In addition, although the built environments that we have investigated do include some “natural” elements (e.g., trees), whether our findings can be generalized to all-nature or quasi-nature (e.g., a designed garden) environments remains a question to be tested in future research. Finally, we focused on quantitative approaches to measure participants’ emotional responses to the urban streets by using visual pattern indicators and bio-monitoring sensors. Qualitative methods such as in-depth interviews could be applied in our future studies as another effective approach to better understand participant’s direct feelings and possible reasons.

## Figures and Tables

**Figure 1 ijerph-18-09677-f001:**
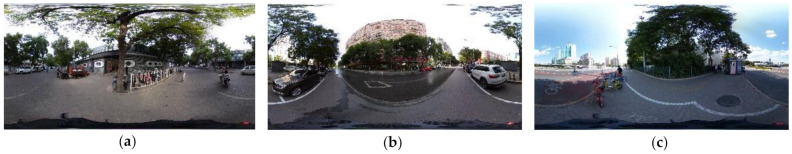
Examples of the street scene panoramic images: (**a**) small street scale; (**b**) medium street scale; (**c**) large street scale.

**Figure 2 ijerph-18-09677-f002:**
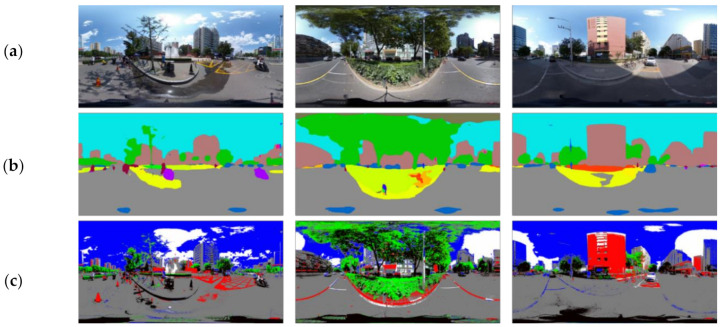
Examples of the street scene semantic segmentation and color classification results: (**a**) original images; (**b**) semantic segmentation results; (**c**) color classification results.

**Figure 3 ijerph-18-09677-f003:**
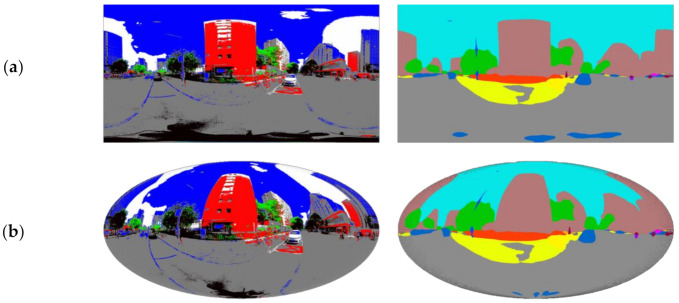
Examples of the Hammer Projection transformation results: (**a**) panoramic images; (**b**) projection transformation results.

**Figure 4 ijerph-18-09677-f004:**
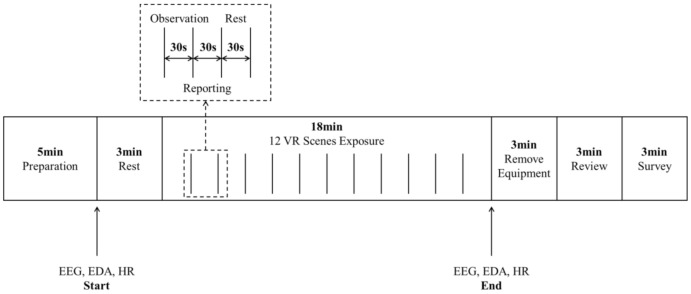
Experiment procedure.

**Figure 5 ijerph-18-09677-f005:**
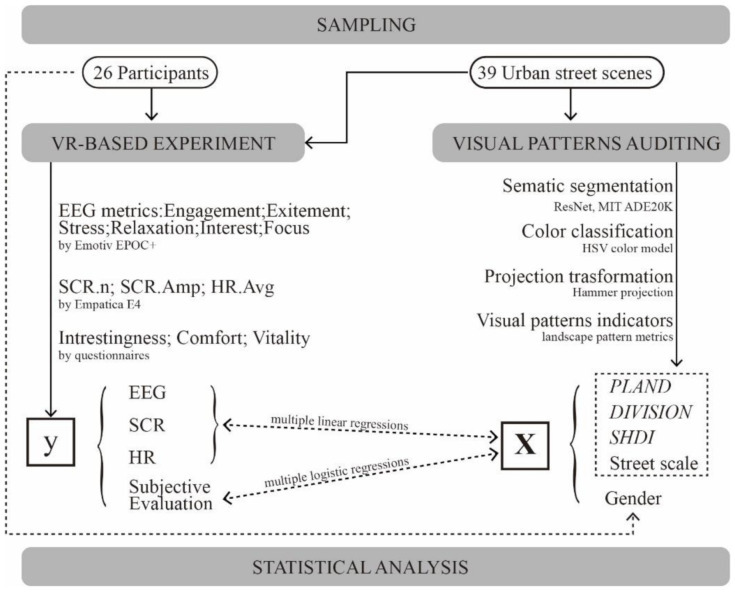
The analytical flowchart. **Notes:** EEG: electroencephalography; SCR: skin conductance responses; HR: heart rate; PLAND: percentage of landscape; DIVISION: landscape division index; SHDI: Shannon’s diversity index.

**Table 1 ijerph-18-09677-t001:** Basic information of landscape pattern metrics in this study. Source: [49].

Level	Name	Calculation Formula		Range
Class (Street scene element group) metrics	Percentage of Landscape	$\mathrm{PLAND}=P_{i}=\frac{\sum_{j=1}^{n}a_{ij}}{A}\left(100\right)$	$P_{i}\;=\mathrm{proportion}\;\mathrm{of}\;\mathrm{the}\;\mathrm{landscape}\;\mathrm{occupied}\;\mathrm{by}\;\mathrm{class}\;i.$ $a_{ij}=\mathrm{area}\;(m^{2})\;\mathrm{of}\;\mathrm{patch}\;ij.$ *A* = total landscape area (m^2^).	(0,100]
Landscape (Entire street scene) metrics	Landscape Division Index	$\mathrm{DIVISION}=1-\overset{m}{\underset{i=1}{\sum }}\overset{n}{\underset{j=1}{\sum }}{\left(\frac{a_{ij}}{A}\right)}^{2}$		[0,1)
	Shannon’s Diversity Index	$\mathrm{SHDI}=-\overset{m}{\underset{i=1}{\sum }}\left(P_{i}\times lnP_{i}\right)$		[0,+∞)

**Table 2 ijerph-18-09677-t002:** Categories of subjective evaluation indicators and the respective scores.

Indicator	Categories		
Interestingness	Insipid	Impassive	Interesting
Comfort	Depressive	Impassive	Relaxed
Vitality	Dull	Impassive	Vibrant
Score	1	2	3

**Table 3 ijerph-18-09677-t003:** Characteristics of the study population, scenes, and outcome measurements.

Category	Variable Name	Mean ± SD or *n* (%)
**Characteristics of the 26 participants**		
**Gender**		
Male	*Gender* (=1)	12 (46)
Female	*Gender* (=0)	14 (54)
**Age**		
18–22	*Age* (=1)	14 (54)
23–25	*Age* (=2)	11 (42)
Older than 25	*Age* (=3)	1 (4)
**Education Background**		
Bachelor Student	*Education* (=1)	12 (46)
Master Student	*Education* (=2)	11 (42)
Doctoral Student	*Education* (=3)	3 (12)
**Characteristics of the 39 scenes**		
**Constituent-element characteristics**		
*Percentage of Landscape* of plant (%)	*PLAND of plant*	16.62 ± 10.05
*Percentage of Landscape* of sky (%)	*PLAND of sky*	18.75 ± 7.16
*Percentage of Landscape* of road (%)	*PLAND of road*	40.09 ± 3.58
*Landscape Division Index* of constituent elements	*DIVISION of CE*	0.80 ± 0.04
*Shannon’s Diversity Index* of constituent elements	*SHDI of CE*	1.64 ± 0.11
**Color characteristics**		
*Percentage of Landscape* of green-class colors (%)	*PLAND of green-class*	5.30 ± 4.74
*Percentage of Landscape* of blue-class colors (%)	*PLAND of blue-class*	15.37 ± 9.67
*Percentage of Landscape* of red-class colors (%)	*PLAND of red-class*	5.75 ± 3.75
*Landscape Division Index* of colors	*DIVISION of CO*	0.79 ± 0.07
*Shannon’s Diversity Index* of colors	*SHDI of CO*	1.40 ± 0.10
**Street scale**		
Large street scale (six lanes or more)	*Large street scale* (=0 or 1)	9 (23)
Medium street scale (four to six lanes)	*Medium street scale* (=0 or 1)	13 (33)
Small street scale (two lanes or less)	*Small street scale* (=0 or 1)	17 (44)
**Characteristics of outcome measurements**		
**Electroencephalography (EEG) (*N* = 208)**		
Engagement	*EEG.En*	0.69 ± 0.09
Excitement	*EEG.Ex*	0.51 ± 0.13
Stress	*EEG.St*	0.67 ± 0.18
Relaxation	*EEG.Re*	0.61 ± 0.15
Interest	*EEG.In*	0.67 ± 0.12
Focus	*EEG.Fo*	0.57 ± 0.16
**Skin conductance responses (SCR) (*N* = 213)**		
Number of SCRs	*SCR.n*	8.12 ± 4.10
Sum of SCR amplitude (mμS)	*SCR.Amp*	1.87 ± 1.82
**Heart rate (HR) (*N* = 310)**		
Average of heart rate (1/min)	*HR.Avg*	76.58 ± 10.88
**Subjective evaluation (*N* = 310)**		
Interestingness	*EVA.In* (=1, 2, or 3)	1.6 ± 0.8
Comfort	*EVA.Co* (=1, 2, or 3)	2.0 ± 0.8
Vitality	*EVA.Vi* (=1, 2, or 3)	1.8 ± 0.7

**Notes:** EEG: electroencephalography; SCR: skin conductance responses; HR: heart rate; PLAND: percentage of landscape; DIVISION: landscape division index; SHDI: Shannon’s diversity index.

**Table 4 ijerph-18-09677-t004:** Results of the linear regressions of the constituent-element characteristics on the physiological indicators.

Indicators		*N* = 208						*N* = 213		*N* = 310
		EEG.En	EEG.Ex	EEG.St	EEG.Re	EEG.In	EEG.Fo	SCR.n	SCR.Amp	HR.Avg
**PLAND of plant**	β	0.019	−0.139	0.145	−0.133	−0.021	−0.013	−0.045	0.008	−0.013
**PLAND of sky**	β	0.115	0.109	0.170	−0.229	−0.183	0.000	−0.054	−0.047	−0.034
**PLAND of road**	β	0.003	0.009	0.094	−0.034	−0.023	0.072	−0.063	0.194 **	−0.016
**DIVISION of CE**	β	0.166	−0.046	0.138	−0.296 **	−0.092	−0.067	−0.001	0.140	−0.055
**SHDI of CE**	β	−0.010	−0.063	−0.126	0.184	0.140	0.113	0.023	0.002	0.068
**Large street scale**	β	−0.066	−0.131	−0.092	0.113	0.115	−0.062	0.115	0.118	0.010
**Small street scale**	β	−0.109	−0.093	−0.048	−0.020	−0.053	0.057	0.008	0.048	−0.008
**Gender**	β	0.019	0.153 **	0.071	0.128 *	−0.248 ***	−0.100	0.224 ***	0.242 ***	−0.165***
**Model Sig.**		0.675	0.092 *	0.754	0.203	0.025 **	0.856	0.090 *	0.014 **	0.388

**Notes:** β: standardized coefficients; significance level * < 0.1, ** < 0.05, *** < 0.01.

**Table 5 ijerph-18-09677-t005:** Results of the linear regressions of the color composition characteristics on the physiological indicators.

Indicators		*N* = 208						*N* = 213		*N* = 310
		EEG.En	EEG.Ex	EEG.St	EEG.Re	EEG.In	EEG.Fo	SCR.n	SCR.Amp	HR.Avg
**PLAND of green-class**	β	−0.090	−0.054	−0.064	−0.031	0.061	−0.080	−0.021	0.028	0.054
**PLAND of blue-class**	β	−0.065	0.187 *	−0.064	−0.063	−0.058	−0.030	0.063	−0.027	0.037
**PLAND of red-class**	β	0.023	0.006	−0.059	−0.062	0.040	−0.040	−0.067	0.070	0.004
**DIVISION of CO**	β	0.050	0.004	0.042	−0.018	0.119	0.089	0.150	−0.054	−0.042
**SHDI of CO**	β	−0.080	−0.082	0.023	−0.037	−0.211	−0.111	−0.127	0.015	0.047
**Large street scale**	β	−0.056	−0.115	−0.023	0.044	0.037	−0.075	0.021	0.141	0.004
**Small street scale**	β	−0.167	−0.025	−0.079	−0.084	−0.125	−0.027	0.047	0.029	0.011
**Gender**	β	0.021	0.152 **	0.063	0.146 **	−0.237 ***	−0.090	0.235 ***	0.219 ***	−0.162 ***
**Model Sig.**		0.942	0.108	0.967	0.432	0.041 **	0.805	0.049 **	0.095 *	0.382

**Notes:** β: standardized coefficients; significance level * < 0.1, ** < 0.05, *** < 0.01.

**Table 6 ijerph-18-09677-t006:** Results of the multiple logistic regressions of the constituent-element characteristics on the subjective evaluation indicators.

Indicators		*N* = 310					
		EVA.In = 2	EVA.In = 3	EVA.Co = 2	EVA.Co = 3	EVA.Vi = 2	EVA.Vi = 3
**PLAND of plant**	β	−0.004	−0.012	0.031	0.056 **	0.007	0.003
**PLAND of sky**	β	−0.003	0.022	−0.013	0.026	−0.071 *	−0.012
**PLAND of road**	β	0.055	0.071	0.013	0.034	−0.017	0.179 **
**DIVISION of CE**	β	−4.860	−2.503	−14.607 *	−10.210	−21.769 ***	−10.131
**SHDI of CE**	β	2.710	2.523	5.558 **	3.895	7.179 ***	8.325 **
**Large street scale**	β	−0.564	0.380	0.615	0.924 *	0.837 *	1.026 *
**Small street scale**	β	−0.100	0.369	−0.101	−0.282	−0.201	0.546
**Gender**	β	−0.062	0.338	−0.016	−0.100	−0.026	0.174
**Model Sig.**		0.559		0.000 ***		0.000 ***	

**Notes:** β: standardized coefficients; significance level * < 0.1, ** < 0.05, *** < 0.01.The reference category of the dependent variable is 1.

**Table 7 ijerph-18-09677-t007:** Results of the multiple logistic regressions of the color composition characteristics on the subjective evaluation indicators.

Indicators		*N* = 310					
		EVA.In = 2	EVA.In = 3	EVA.Co = 2	EVA.Co = 3	EVA.Vi = 2	EVA.Vi = 3
**PLAND of green-class**	β	−0.015	−0.007	0.110 **	0.108 **	0.125 ***	0.086
**PLAND of blue-class**	β	0.010	0.054 *	0.029	0.011	0.020	0.080 ***
**PLAND of red-class**	β	0.014	0.000	−0.020	−0.044	−0.001	0.017
**DIVISION of CO**	β	1.677	−1.952	−10.311 **	−8.567	1.692	−6.609
**SHDI of CO**	β	0.049	3.541	7.796 **	6.369	−1.926	4.560
**Large street scale**	β	−0.492	0.425	0.622	1.215 **	0.506	0.836 *
**Small street scale**	β	−0.234	0.694	−0.069	−0.439	−0.345	0.726
**Gender**	β	−0.034	0.344	0.056	−0.053	0.083	0.171
**Model Sig.**		0.477		0.000 ***		0.000 ***	

**Notes:** β: standardized coefficients; significance level * < 0.1, ** < 0.05, *** < 0.01.The reference category of the dependent variable is 1.

**Table 8 ijerph-18-09677-t008:** The correlation coefficient of the multi-dimensional perception indicators.

Indicators	EEG.En	EEG.Ex	EEG.St	EEG.Re	EEG.In	EEG.Fo	SCR.n	SCR. Amp	HR.Avg	EVA.In	EVA.Co	EVA.Vi
**EEG.En**	1											
**EEG.Ex**	0.079	1										
**EEG.St**	0.220 ***	0.160 **	1									
**EEG.Re**	0.097	0.076	0.258 ***	1								
**EEG.In**	0.227 ***	−00.075	0.308 ***	0.194 ***	1							
**EEG.Fo**	0.175 **	0.247 ***	0.257 ***	−0.008	0.400 ***	1						
**SCR.n**	0.031	0.161 **	0.120	0.039	0.023	0.192 **	1					
**SCR.Amp**	0.095	−0.014	0.085	−0.118	−0.046	0.090	0.411 ***	1				
**HR.Avg**	0.156 **	0.086	0.080	0.082	0.026	0.052	−0.181 ***	0.141 **	1			
**EVA.In**	0.038	−0.058	0.022	0.032	−0.034	0.014	−0.073	0.028	0.037	1		
**EVA.Co**	−0.057	−0.198 ***	−0.018	0.088	−0.062	−0.067	−0.112	−0.061	−0.010	0.273 ***	1	
**EVA.Vi**	−0.058	−0.179 ***	−0.037	0.082	−0.049	−0.040	−0.059	−0.025	0.043	0.436 ***	0.561 ***	1

**Notes:** significance level * < 0.1, ** < 0.05, *** < 0.01.

## Data Availability

Data available upon request due to privacy/ethical restrictions. The data presented in this study are available upon request from the corresponding author.

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
