# Peer review of "Emotional Responses to the Visual Patterns of Urban Streets: Evidence from Physiological and Subjective Indicators"

_ijerph, 2021, doi:10.3390/ijerph18189677_

Round 1
Reviewer 1 Report
This paper demonstrates citizen's emotional response to various urban street scenes through a VR-based immersive exposure experiment considering both objective and subjective evaluation. However, this paper needs to be revised in more evident conclusions of their findings.
Major comments:
1. Regarding indicators ( EEG, EDA, and HR / interestingness, comfort and vitality), add more explanations.
- 1.1 [line 15~] I would like to recommend not to use abbreviations on the abstract part.
- 1.2 [line 115~] Provide more explanation on why these indicators are used as indicators for physical monitoring.
- 1.3 For the subjective indicators, interestingness, comfort and vitality were used. Please add more descriptions (e.g., why you chose these indicators, what explanation we can get from these indicators.).
2. Provide more evidence on "[line 24~25] combining the traditional, interview-or questionnaire-based research techniques would help better unveil the mechanisms of the environment-emotion relationship". In the manuscript, this conclusion was not fully discussed.
3. [line 32~] "a self-evident fact that justifies the significance of the subject" is lacking evidence (leaps of logic).
4. I would like to recommend that the introduction part should be divided into the introduction and literature review.
5. [line 49~] Describe more on "exposure-cognition-behavior response framework".
6. [line 95~] In the paper, you mentioned that accurate comprehensive measurements of the environment are limited in the previous studies. In other words, your framework would overcome this limitation. Please provide more discussion on why your methodology is comprehensive than the previous works of literature.
7. In the study, many procedures and methodologies were used. Please add some analytical flowchart of the whole process in the Method section.
8. In this study, 26 students of Peking University participated in the experiment, which might bring an issue on the representation of sampling.
- 8.1 Provide more detailed information on why you choose these participants as a sample for representing your experiment.
- 8.2 Provide general information of the participants (age, gender, health status, etc.)
- 8.3 This limitation should be discussed in the conclusion section.
Minor comments:
1. I would like to recommend adding limitations and future works of this study in the conclusion section.
2. Academic writing:
- There is no consistency in the use of capital letters and quotation marks.
- Leaps of logic.
- Hard to read sentences.
Author Response
Dear Reviewer:
Thank you very much for your careful review of our manuscript. We appreciate very much your thoughtful comments, which we think would indeed help us significantly improve the quality of the manuscript toward a more consistent, clear, and academically-rigorous one. Per your comments, we have carefully revised the manuscript. Please kindly refer to the attached file for details of our response to your comments.
Many thanks again
The authors

Reviewer 2 Report
Given the increasing rates of urbanization worldwide, this paper provides a timely contribution in understanding the psychosocial effects of the built environment on well-being. The researchers adopt a novel, quantitative approach using VR technology to collect data, which, despite the potential limitations of distortion, seems to yield some interesting findings. Most notably, the notion that a fragmented landscape engenders negative emotions for citizens.
While my comments are largely supportive, there was scope to improve this paper overall. Please consider the following points;
- The Introduction provides sound context for the research, however, some assertions require further justification. For example, the opening line (Line 30) needs to be supported by the relevant literature (i.e. what 'long-established intellectual tradition'? Who are the seminal authors in this space?). Similarly, Lines 30-36 need to be further justified or rewritten. The mere notion that people live in cities, doesn't justify the 'long-established intellectual tradition'.
- I appreciate that the authors are not likely writing in their native language, however, grammar and written expression overall requires attention. For example, there are a number of places where incorrect words are used (i.e. Line 36 - do the authors mean 'sensory', not 'sensuous'?; Line 57 - 'restoration' does not seem to be the appropriate word; Line 492 - 'uncomfortness' should be replaced with 'discomfort'). There are several instances where meaning also becomes unclear due to long sentences (see for example Lines 42-47).
- Planer photos (Line 83) needs to be defined.
- I appreciate the approach taken by the authors to focus on the built environment (as opposed to the abundance of research on the impacts of greenspace), however, the authors should also acknowledge that emotive experiences (in line with Gestalt Psychological approaches) also involve other senses, including olfactory and auditory.
- Line 57-58 - define a 'Healing Landscape' and return to this idea in the Discussion section.
- Lines 413-414: the assertion made here is unclear. Perhaps better define 'fragmentation' in the Introduction? What is the relationship between fragmentation and diversity?
- Line 457 - does uniformity apply to vegetation as well, or just built structures?
- Overall, the authors need to strengthen their Results/Discussion section by linking back to the literature discussed in preceding sections. There is also an opportunity to add some discussion around the semiotics of architecture (i.e. after Saussure and Eco's work). This would provide a more nuanced and richer understanding of the qualitative dimensions to the emotive responses to urban streetscapes. Relying solely on quantitative data and analysis limits understanding of such a highly subjective and emotive research question. As such, I strongly encourage the authors to include some of this seminal literature and at the very least, acknowledge the limitations of a quantitative approach to this research question.
Author Response

(The authors gave the same response as above.)

Round 2
Reviewer 1 Report
The authors have addressed most of the reviewer's comments to a sufficient extent.
However, I cannot fully agree with your revised "limitation and future work" section.
You have listed three limitations. Based on your limitations, does "How to interpret the thousands-year-old, rich design wisdom of the architects with global measurements of the physical environment remains a subject for future inquiries." indeed indicate your future work according to the limitations? What about suggesting each future work based on each limitation or adding more persuasive explanations?
Also, please consider moving this content to the conclusion part. The added sub-section "4.4 limitation and future work" seems not look like accompanying with other sub-sections (4.1, 4.2, and 4.3) in the discussion part.
Author Response
Dear Reviewer #1: Thank you very much for your careful review of our manuscript. We appreciate very much your comments on the “Limitations and Future Work” part, and also your suggestion on moving the part to the conclusion part. Per your suggestions, we have moved the “Limitations and Future Work” part to the conclusion section. Also, we have removed the vague expression about the conventional design wisdom. The part now contains three specific points of limitations, each corresponding to one possible direction of future work. We hope these revisions would help clarify any confusion here. Please kindly refer to the revised manuscript for details.Reviewer 2 Report
I commend the authors for attending to the suggested revisions . There is now a much clearer and consistent narrative throughout the paper. I also applaud the authors' revisions of the Methods section (this is now appropriately detailed) and the acknowledgement of the research limitations. Apart from some minor language/style errors listed below, I support this paper's publication;
Please revise the following;
Figure 5 - The grey box STATISTIC ALANALYSIS is misspelled. This should change to: STATISTICAL ANALYSIS
Table 3 - Under 'Education Background' - DOCTOR STUDENT should change to DOCTORAL STUDENT
Lines 2213-2214 "Besides possible technical errors, why most of the “common” measurements of the EEG data are found not being able to be related to people’s expressed emotions?" is unclear. Do the authors mean?: Notwithstanding possible technical errors, questions remain as to why most of the "common" EEG data measurements do not correlate to the respondents' expressed emotions.
Author Response
Dear Reviewer #2: Thank you very much for your approval of our revised manuscript, and also for your additional suggestions on typo corrections and the language style. We have revised all the corresponding parts per your suggestions. Please kindly refer to the revised manuscript for details.